# Cis-Regulatory Elements in Mammals

**DOI:** 10.3390/ijms25010343

**Published:** 2023-12-26

**Authors:** Xingyu Liu, Mengjie Chen, Xiuwen Qu, Wenjing Liu, Yuting Dou, Qingyou Liu, Deshun Shi, Mingsheng Jiang, Hui Li

**Affiliations:** State Key Laboratory for Conservation and Utilization of Subtropical Agro-Bioresources, Guangxi Key Laboratory of Animal Breeding, Disease Control and Prevention, College of Animal Science and Technology, Guangxi University, Nanning 530005, China

**Keywords:** cis-regulatory elements, enhancers, promoters, mammals

## Abstract

In cis-regulatory elements, enhancers and promoters with complex molecular interactions are used to coordinate gene transcription through physical proximity and chemical modifications. These processes subsequently influence the phenotypic characteristics of an organism. An in-depth exploration of enhancers and promoters can substantially enhance our understanding of gene regulatory networks, shedding new light on mammalian development, evolution and disease pathways. In this review, we provide a comprehensive overview of the intrinsic structural attributes, detection methodologies as well as the operational mechanisms of enhancers and promoters, coupled with the relevant novel and innovative investigative techniques used to explore their actions. We further elucidated the state-of-the-art research on the roles of enhancers and promoters in the realms of mammalian development, evolution and disease, and we conclude with forward-looking insights into prospective research avenues.

## 1. Introduction

Unveiling the inherent laws of life is one of the main research objectives of modern biological science, and deciphering the regulatory mechanisms of phenotypes to investigate their associations with animal development, evolution and disease is of paramount importance in contemporary biological research directions. Cis-regulatory elements (CREs), comprising enhancers and promoters (EPs) and based on their chromatin three-dimensional spatial structure, regulate the precise initiation and efficiency of gene transcription by binding with transcription factors (TFs). These play an indispensable role in the regulation of gene expression. Within organisms, development is a highly ordered and dynamic process, involving the temporal and spatial regulation of numerous genes. EPs can interact with regulatory factors, orchestrating gene expression across various developmental stages. EPs have also played pivotal roles during mammalian evolution. By comparing EP sequences from different mammals, conserved and variable regions in these sequences have been identified, further shedding light on the evolutionary dynamics of mammalian gene regulatory networks and uncovering the genetic differences between mammalian species. With respect to diseases, EPs have been associated with cancers, neurological disorders and autoimmune diseases. Exposing their roles in disease mechanisms promises to offer new molecular targets for the development of innovative therapeutic methods. This article summarizes research advances on the concept of EPs and their roles in mammalian development, evolution and diseases. In addition, we elucidate the latest research perspectives, highlighting challenges that exist, with the aim to inspire new insights and directions for future EP studies.

## 2. Cis-Regulation Element

Enhancers are short DNA sequences within the genome that, by binding with various TF and cofactors, act upon promoters to regulate gene transcription. Enhancers operate without being limited by either direction or distance and are not restricted to the regulation of any specific gene. A single enhancer, influenced by the three-dimensional conformation of chromatin, often regulates the transcription of multiple genes [1].

In mammalian cells, active enhancer DNA sequences are loose and open, and they provide the environmental conditions for enhancers to transcribe genes [2]. Studies have shown that RNA polymerase II (Pol II) acts on promoters while also inducing the transcription of enhancers, resulting in the production of enhancer RNA (eRNA) [3]. ERNAs usually have short half-lives, low abundance and they lack RNA processing ability. They were initially regarded as sequencing noise and a by-product of transcription [4]. In mammals, the C-terminal domain of the largest subunit of Pol II contains Ser5, which can be in the phosphorylated state of Ser5 and subsequently initiate transcription [5]. In addition, phospho-Pol II (Ser5P) has been shown to be enriched in enhancers [6]. The insertion of transcription terminators into the globin genes enhancer (*HS2* enhancer) stops the formation of eRNA [7]. Interference with eRNAs with small interfering and short hairpin RNAs reduces the transcription of its enhancer target genes [8]. This evidence not only robustly confirms the existence of eRNAs but also signifies their crucial role in the regulation of enhancer target genes. Hence, active enhancers can transcribe and generate eRNAs (Figure 1) and may regulate the transcription of target genes through eRNAs.

A promoter is a DNA sequence that allows a specific RNA transcript to be transcribed. The structure of the promoter affects its affinity for binding with RNA polymerase, thus affecting transcription levels. The structure of eukaryotic promoters typically consists of three parts: the core, proximal and distal regions [9]. Sequence-wise, the core promoter sequence contains the transcription start site (TSS). The vicinity of the TSS usually binds with general transcription factors (GTFs), Pol II and the mediator, leading to the formation of the pre-initiation complex (PIC). This region is termed the “core promoter” area, usually residing within 50 bp upstream and downstream of the TSS. It serves to determine the precise location and direction of transcription [10].

Core promoters are not structurally fixed, but classical elements include the TFIIB recognition element (BRE), TATA box, initiator (INR) and downstream promoter element (DPE). The BRE often accompanies the TATA box, and it is located either upstream or downstream of this and can significantly influence the binding of RNA polymerase to it [11]. The TATA box usually appears approximately 25 bp upstream of the initiation of transcription and it determines the onset of this process. It is also a binding site for RNA polymerase [12]. The INR element, encompassing the transcription initiation site, is a common functional element in the promoter of many protein-coding genes [13], playing a vital role in the initiation of transcription [14]. Moreover, core promoters containing the DPE element often lack the TATA box element [15]. The function of the DPE is similar to that of the TATA box and when it is mutated, it directly affects gene transcription [16]. The complexity of promoter sequences mostly lies in the core promoter, potentially due to the diverse and interchangeable GTFs that bind with it. These enable the precise recognition of various core promoter sequences [17]. The surrounding sequences of the core promoter also bind with TFs, recruiting cofactors that impact on the pre-initiation complex bound to the core promoter, which subsequently influence the initiation and elongation of transcription [18]. The regions around the core promoter that can bind to TFs, and based on the proximity to the gene, are termed the “proximal promoter” and “distal promoter” areas, which are collectively referred to as the “promoter” (Figure 2) [19].

It should be mentioned that the TSS of the majority of genes are in pairs. Thus, there is another TSS in the proximal upstream region of the opposite strand. This implies that promoters can undergo “divergent transcription”, producing promoter upstream transcripts (PROMPTs)/upstream antisense RNAs (uaRNAs) and mRNAs. PROMPTs are typically shorter than 500 bp, are not usually spliced and can be readily degraded by exosomes [20].

## 3. Detection of CREs on the Genome

The rapid development of high-throughput sequencing technology has made it possible to detect CREs in a genome-wide manner with high efficiency (Table 1). The current identification methods for detecting EPs are broadly divided into the following six categories.

### 3.1. Recognition Based on Epigenetic Modifications

Numerous studies have shown that histone acetylation is mainly associated with gene activation. However, methylation is associated with repression as well as activation depending on its location and status [35].

EPs undergo histone modifications of their chromatin. Enhancers undergo cofactor-mediated acetylation of lysine 27 on histone H3 (H3K27ac) and mono-methylation of lysine 4 on histone H3 (H3K4me1) [36]. Depending on the histone modifications, enhancers are classified based on their genome into three states: initial, accumulation and activation. Enhancers in the initial state are only modified by H3K4me1, which is inactive. The ones in the accumulation state are enriched with H3K4me1 as well as tri-methylation of lysine 4 on histone H3 (H3K4me3) simultaneously, which is at the threshold between inactive and active; they are pending activation and are not yet functional. The enhancers in the activation state already have both H3K4me1 and H3K27ac modifications, and these play a role in the initiation of gene transcription [36].

The promoter states are only categorized into resting and activated ones. Because the level of enrichment of H3K4me3 in the vicinity of gene promoters is positively correlated with the level of gene expression, this parameter is usually used as a marker histone for active promoter recognition [37]. Depending on the different histone modifications, genome-wide detection of EPs can be achieved using high-throughput sequencing technologies such as ChIP-seq [21], CUT&RUN [22] and CUT&Tag [22] (Table 1).

### 3.2. Recognition Based on TFs and Cofactors

ChIA-PET assays have shown that the percentage of EP interactions mediated by the widely expressed transcription factor, Yin Yang 1 (YY1), was 27%, and the percentage of inter-insulator interactions was 1% [38]. When YY1 was knocked down, it affected the formation of chromatin loops, which resulted in a significant decrease in gene expression levels [39]. This suggests that YY1 is a key factor mediating EP interactions. P300 protein acts as a cofactor with histone acetyltransferase activity. Therefore, p300 can change the openness of chromatin and facilitate the entry and binding of transcription factors. Studies have shown that p300 has an extremely strong role in predicting the presence of enhancers, and more than 80% of the enhancers in the human and mouse genomes can be predicted by its presence [40]. Therefore, by combining the roles of YY1 and p300 in mediating EP interactions, the presence of EPs can be predicted by using ChIP-seq, CUT&RUN and CUT&Tag technologies (Table 1).

### 3.3. Recognition Based on Chromatin Accessibility (CA) Regions

In the genome, some DNA regions are naked and exist without nucleosomes, and these are known as CA regions. CA is the physical basis for DNA to be able to bind to regulatory factors such as TFs, which is a necessary condition for gene transcription [41]. EPs regulate gene transcription by binding to TFs. Therefore, EPs can be predicted based on the presence of CA regions and the current techniques for their detection are DNase-seq [24], MNase-seq [25], FAIRE-seq [26] and ATAC-seq [27] (Table 1).

### 3.4. Recognition Based on Comparative Genomics

With the development of mammalian genome sequencing and comparative genomics, a steady stream of conserved elements has been deduced. There are a large number of CREs in these conserved elements, and this is an effective way to find and identify enhancers. Pennacchio et al. confirmed that 45% of the 167 conserved elements obtained from comparative analyses of human, mouse and rat genomes have enhancer functions [42]. Xu et al., through sequence conservation analysis and histone modification detection, identified the enhancer element for PPARγ2 [43]. Xin et al. compared and analyzed the conserved sequences in the Zokor genome and mouse genome with ENCODE data, and then they identified a large number of CREs [44]. We have summarized the conserved elements that have been deduced in different phyla of mammals [45,46,47], with the expectation of providing a usable resource for CRE identification (Table 2).

### 3.5. Recognition Based on Transcription Products

Most active EPs can be transcribed to produce non-coding RNAs (eRNAs and PROMPTs). ERNAs and PROMPTs are short and are typically less than 200 bp. Usually, their transcription starts at the edge of the nucleosome deletion region and it is bidirectional [4]. ERNAs and PROMPTs are usually not polyadenylated and sheared, so they are highly susceptible to degradation [19]. Therefore, it is difficult to detect eRNAs and PROMPTs in the RNA-seq data commonly measured in studies. Transcript sequencing technologies such as GRO-seq [28], PRO-seq [29], CAGE-seq [30] and Start-seq [31] possess much higher sensitivities and can be used to identify eRNAs and PROMPTs which can subsequently predict the occurrence of EPs [48].

### 3.6. Identification Based on CRE Activity

Among the methods to identify regulatory elements based on activity, the pGL3 luciferase reporter system is currently a more accurate way to identify EPs [49], but it has only been adapted to the identification of individual sequences and it is not applicable to large-scale assays [50]. Therefore, based on the above approaches, more techniques have been derived for the identification of active EPs using high-throughput technologies. These include, for example, SIF-seq [32], MPRA [51] and STARR-seq [52] (Table 1).

## 4. Experimental Techniques for Assessment of CREs

Activity characterization is one of the most important aspects in the assessment of CREs. The dual luciferase reporter system is currently a common and effective way to measure the activity of CREs. It is based on the enzymatic oxidation of luciferin to oxyluciferin by luciferase, which then emits a quantifiable amount of bioluminescence. Ultimately, the activity of the regulatory elements is detected based on the level of fluorescence intensity [53]. This technique is sensitive and flexible, and different vectors can be constructed for different types of regulatory elements in order for them to be measured.

In addition, gene editing technology has been widely used in verifying the functions of CREs. CRISPR/Cas9 extremely derivative technology is currently the most applied technique in the study of CREs. CRISPR/Cas9 technology is easy to operate and efficient, and it can be applied to explore the function of CREs on individuals [54]. CRISPR/dCas9 was created when mutations were introduced at the nuclease site of Cas9 on the basis of CRISPR/Cas9 [55]. However, CRISPR/dCas9 no longer cleaves DNA but binds directly to specific DNA sites under the guidance of gRNA and does not irreversibly affect the target site. This feature was exploited to create the CRISPRa technology by fusing dCas9 with transcriptional activators, such as VP64 [56]. By designing different gRNAs to target CRE sites, CRISPRa can activate regulatory elements and regulate the transcription of genes. Similarly, the CRISPRi system was formed by fusing dCas9 with the transcriptional repressor structural domain KRAB, which avoids a drawback of the dCas9 system which has low repression efficiency in mammals. This technique is able to effectively repress the regulatory elements in order to fulfill their own roles [57]. The CRISPRa and CRISPRi technologies play complementary roles in the identification and investigation of the functional mechanisms of CREs. In addition, they have the advantages of a low off-target rate and high efficiency and have become the mainstream technology for the study of CREs.

## 5. Similarities and Differences between EPs

With increasing knowledge about CREs, the definition of EPs has now become controversial, mainly focusing on whether these molecules are two completely independent regulatory elements. Initially, enhancers were defined as regulatory elements that promote the expression of target genes [58] and promoters as those that initiate target gene transcription [59]. Based on these definitions, the reader relied on the differences between EPs for differentiation. Thus, differences in histone modifications, sequence CG content and the number of transcription products (eRNAs vs. PROMPTs) between EPs are the current basis for distinguishing these molecules. However, in recent years, it has been found that the above definitions of EPs do not support many of the current findings [60]. This is related to the existence of certain similarities between EPs. In terms of the chromatin background, both EPs with biological activities are located in the open chromatin ultrastructure [41]. In terms of sequence structure, enhancer sequences are simply designed to recruit TFs to promote transcription from distal promoters. But the discovery of eRNAs is at odds with this perception, suggesting that the enhancer recruits TFs for its own transcription, and core promoter-like elements have been found in enhancers, and these are structurally identical to the promoters [61]. In terms of transcription products, both eRNAs and PROMPTs are produced by divergent transcription and have shorter lengths and are not sheared and readily degraded. With respect to transcription factors and cofactors, Ser5-phosphorylated Pol II is enriched in EPs, and both EPs are able to recruit cofactors, p300 and CBP [62].

Thus, the existence of many similarities between EPs in terms of chromatin background, sequence structure, transcription products, TFs and cofactors have called into question whether EPs are two completely independent regulatory elements. In addition, many studies have reported that many enhancers function as promoters and vice versa [63]. There is currently a great deal of controversy regarding the actual boundary between EPs. For the differences between EPs, they can basically be explained by differences in the strength of core sequence activity. Therefore, it is worthwhile to explore in more detail whether EPs are two completely independent regulatory elements as defined.

## 6. Molecular Mechanism of EP Interactions

Studies have shown that EPs are capable of interacting at a distance within the genome, and the mechanisms by which they do so have generated great interest. With the development of CA detection technology, chromatin conformation capture technology [64], RNA in situ conformation sequencing [65] and other related technologies, it is possible to study these interactions in depth (Figure 3). It was previously hypothesized that chromatin formed a ring-like structure that allowed the enhancer to interact with the promoter by making physical contact and this was confirmed by electron microscopy [66]. This type of chromatin conformation was called “chromatin loops”, and these can be broadly divided into two categories: one is “TAD” and the other is a more subtle division, which we call “loop”.

TAD is thought to be formed and mediated mainly by CTCF and adhesion [67]. The model consists of adhesion proteins recruiting NIPBL to form an adhesion protein complex with the MAU2 protein and extruding the chromatin structure outward until it encounters two reverse binding CTCF blocks. These are eventually released from the chromatin by the WAPL protein [68]. The current study shows that disruption of CTCF/cohesin results in a slight genome-wide decrease in transcript levels, a substantial decrease in transcript levels for a small number of genes as well as a reduction in long-range interaction sites [69]. Many of the down-regulated genes contain CTCF sites proximal to the promoter, while their cognate enhancers are usually located distally [70]. In addition, chromatin conformation capture (3C) and fluorescence in situ hybridization (FISH) techniques have confirmed the high frequency of chromatin interactions within TAD [71]. When the CTCF binding site is disrupted, some of the enhancers interact with promoters outside the TAD range [72]. This suggests that the CTCF/cohesin-mediated TAD loop can mediate interactions between distal EPs and can restrict the range of action of enhancers within the TAD.

Loop is a finer interaction structure formed with an increase in the TAD identification depth, which is due to the physical spatial contact between EPs during TAD formation. The stabilizing interactions of loop are generated by the action of TFs and their cofactors. Thus, the loop is a much smaller chromatin conformation present in the TAD and it is the primary structure which mediates the EP interactions [73]. YY1 is a transcription factor widely found in mammalian cells that binds hypo-methylated DNA sequences to form homodimers, thereby facilitating the formation of loops. As mentioned above, YY1-mediated EP interactions account for 27% of interactions in the whole genome. When YY1 is deleted, the expression levels of genes across the genome are significantly reduced. Therefore, it is now widely recognized that YY1 is an important factor which mediates the formation of loops (Figure 4) [74].

The presence of chromatin loops provides spatial possibilities for EP interactions. However, the transcription levels of genes in chromatin loops are not the same, even in the same chromatin loops for neighboring genes [75]. This suggests that EP interactions are independent of position within the chromatin loop and that they may be subjected to targeted regulation based on spatial conformation. It was found that the promoter proximal tethering element (PTE) may mediate enhancer-targeted promoter interactions [75]. Interestingly, similar sequences can normally be found within or near enhancers. For the mechanism of how PTEs function, it has been suggested that PTEs may exert targeting effects by regulating the chromatin openness of promoters [76]. Chromatin accessibility currently explains more of the phenomenon of EP targeting, and many studies exist to confirm that enhancers target distal promoters across proximal promoters because of their effects on the accessibility of chromatin [77]. In addition, the differential expression of homologous genes which should be co-expressed at different times can also be explained by CA.

However, on the basis of the same chromatin loop with the same chromatin openings, there are still differential expression levels between proximate genes that cannot be explained by the above remarks [78]. In *Drosophila*, the *Oaf* promoter is closer to the *DPP* enhancer, but the *Oaf* is not regulated by the *DPP* enhancer. However, when the core promoter of Oaf is replaced with HSP70, *Oaf* is then regulated by the *DPP* enhancer [79]. This suggests that the core promoter sequence may be an important factor in EP targeting. In this regard, by classifying the core promoter sequences of different types of genes, the core promoter elements of developmental genes have been found to differ from those of housekeeping genes [11]. This suggests that different core promoter sequences may have different functions and these may be regulated by different enhancers. Similarly, enhancers can be divided into those for developmental genes and others for housekeeping genes. Both types of enhancers are enriched with specific TF binding motifs. For example, developmental gene enhancers are enriched for Trl and housekeeping enhancers are enriched for DREF [80]. Activator bypass experiments targeting TFs have revealed that these bind differentially to housekeeping and developmental enhancers: Trl is recruited at the developmental core promoter and DREF is recruited at the housekeeping core promoter [81]. This suggests that the targeted interactions of different EPs with promoters and their transcriptional activation may be mediated by different TFs.

TFs recruit cofactors when exerting their own effects, and these can work on a feedback mechanism to the DNA-binding activity of TFs [82]. Similar to TFs, cofactors exhibit core promoter preferences [83]. Developmental core promoters are commonly activated by Mediator and CBP/P300, and housekeeping core promoters are often activated by Chromator and TFIID subunit TBP-associated factor 4 (TAF4) [84]. Therefore, the targeted activation of enhancers for promoters is highly dependent on cofactors. It is traditionally believed that cofactors, such as Mediator, are able to connect promoter PIC and enhancers by means of physical linkage, thereby realizing enhancer-to-promoter communication [85]. However, with the application of cryo-electron microscopy (Cryo-EM), this assumption was broken [86,87]. Cofactors were found to often have enzymatic activity and they can modify many proteins. TFs, Pol II and cofactors are regulated by post-translational modifications (PTMs). For example, Pol II requires a cofactor, p300, to acetylate PTMs in order to be active on growth factor-responsive genes [88]. Thus, PTMs may form a communication link between EPs.

The roles of eRNAs and PROMPTs in EP interactions have also been gradually revealed. In macrophages, when the eRNAs associated with the *MMP9* and *CCX3CR1* enhancers were inhibited by siRNAs, the transcript levels of these genes were markedly reduced, but there was no effect on the transcript levels of the genes closest to the eRNAs [8]. This suggests that eRNAs may mediate the targeting recognition of EPs [89]. Recently, based on RIC-seq technology, researchers found that eRNA and PROMPT interaction sites were highly enriched in Alu sequences and these may form RNA duplexes in a reverse complementary manner in order to enable promoter–enhancer targeted interactions [90]. Moreover, upon the insertion of Alu sequences into promoter regions that do not normally contain such sequences, they were able to be activated by enhancers containing Alu elements. In addition, a significant increase in the level of transcription of otherwise unregulated target genes and a significant increase in the frequency of spatial contacts were seen. This not only showed that eRNAs and PROMPTs have important roles in EP targeting interactions, but also suggests that genomic repetitive sequences are important for this to occur.

In summary, in order to illustrate the molecular mechanisms of EP interactions, here we looked at the chromatin spatial conformation formed by EPs and showed how they achieved spatial proximity at a distal region to where they are found. On the basis of chromatin spatial conformation, the possible reasons for EP targeting interactions were elaborated and summarized through four aspects, namely, CA, TFs, cofactors and eRNAs.

## 7. Advances in CREs in Mammals

### 7.1. CREs and Mammalian Development

Enhancers play a decisive role in the development of certain traits. The *Shh* gene plays an essential role in the development of the distal parts of the skeleton, and its expression is regulated by the key enhancer ZRS, which markedly promotes this process by recruiting the transcription factor ETS1. In a constructed mouse model, the knockdown of ZRS resulted in a severe decrease in Shh expression levels and the absence of distal bones in the mouse [91]. This suggests that ZRS has an extremely important role in the development of the distal limbs. A single enhancer can even play a decisive role in individual traits.

It is well known that the Y chromosome is necessary for the development of males, the expression of the sex-determining gene, *Sox9*, cannot be separated from the regulation of the Sry protein on the Y chromosome, and mammals develop male organs when it is present [92]. Gonen et al. identified a direct reciprocal enhancer 13 (Enh13) with Sox9 and when it was knocked down in the XY chromosome of male mouse, and they showed that the expression of Sox9 was significantly reduced and the ovaries as well as other female genitalia features grew in the male mouse [93]. This result directly illustrates that individual enhancers may also have a strong regulatory capacity and can directly play a decisive role in individual traits.

There is a complex synergy of enhancers that regulate the development of traits. Cochlear hair cells are receptors for sound, and the transcription factor, Atoh1, has long been an important target for the study of these cells. The deletion and overexpression of Atoh1 can directly lead to the loss and increase in cochlear hair cells, respectively [94]. Three enhancers of *Atoh1* (Eh1, Eh2 and Eh3) were found, and by knocking down each of the three enhancers in turn, researchers found that only the reduction in Eh1 had no effect on the development of cochlear hair cells after its deletion [95]. However, when Eh2 and Eh3 were knocked down, a significant reduction in the number of cochlear hair cells was found and after knocking out all three, and these cells were almost completely absent. This suggests that there seems to be a complex synergy between enhancers to regulate the number of cochlear hair cells.

Hoxd is able to control the development of upper limbs/front legs, wrists, fingers, ankles as well as toes [96]. Low or high levels of Hoxd expression result in developmental defects of the mammalian limbs (e.g., either short fingers or polydactyly). Studies have revealed that there are seven enhancers that regulate Hoxd. Because of changes in their chromatin topology, all seven enhancers change their own activities and collaborate with each other to allow Hoxd to be expressed to varying degrees at different developmental periods, which in turn regulates the development of related traits [97].

Enhancers spatiotemporally and temporally regulate gene expression in specific ways, thereby controlling trait development. The transcription factor Pax6 plays an important role in regulating the developmental morphology of the eye. When heterozygous Pax6 is absent in the retina, it leads to iris hypoplasia [98,99]. Bhatia et al. found that the enhancer at 150kb downstream of the *Pax6* promoter, SIMO, interacted with the *Pax6* promoter. When SIMO was knocked down, there was no effect on the initial expression of Pax6, but after a period of time, its expression was significantly reduced [100]. It was also shown that SIMO was not necessary for initial Pax6 expression but that it was essential for the maintenance of its expression. This suggests that enhancer regulation can be time-specific.

The earlier mentioned gene, *Sox9*, is associated with cartilage, cranio-maxillofacial and gonadal development, and it has been shown to be finely regulated in different tissues. Studies have shown that there are enhancer clusters upstream of *Sox9*, and the regions of these enhancers are highly open in cranial neural crest cells. However, the openness is significantly reduced after cranial neural crest cells differentiate into chondrocytes, suggesting that these enhancers may regulate the early growth and differentiation of the neural crest and no longer function in cartilage [101]. When clusters of these enhancers were knocked out in mice, the expression levels of Sox9 were significantly decreased and the animals showed significant jaw hypoplasia, but there was no effect in the development of the forelimb cartilage and no significant change in the expression level of Sox9 [102]. This suggests that there is tissue specificity of enhancers in regulating the development of traits.

### 7.2. CREs and Mammalian Evolution

The enhancers are recognized as an important source of determining species diversity. Bats are mammals with unique flight capabilities and have long forelimbs to accommodate their ability to fly. Prx1 plays an important role in various processes of limb development and it is highly conserved in bats and mice. Behringer et al. analyzed and identified enhancers of the *Prx1* gene in mice and bats and were able to replace the mouse *Prx1* enhancer with that of the bat. The results showed that the forelimbs of the mice grew significantly when compared to the wild type [103]. This suggests that the *Prx1* enhancer may be responsible for the unique flight ability of bats.

Most mammals have evolved with a social hierarchy. To explore the evolutionary mechanism of social hierarchy, Wang et al. identified PAS1 as an enhancer of the gene *Lhx2*, a transcription factor involved in brain development. When the PAS1 molecules were isolated from different mammals for in vitro activity validation, they were found to exhibit different cis-regulatory activities and had different effects on Lhx2 expression in the mouse embryonic nervous system. Further studies found that the PAS1 knockout mouse lacked social stratification. A PAS1 knock-in mouse model showed that PAS1 determined social dominance and subordination in adult mice and that social status could even be subverted by mutating these molecules [104]. This suggests that PAS1 was involved in the formation of social hierarchies in mammals. The ruminant multi-chambered stomach is a key mammalian organ innovation. Pan et al. combined ATAC-seq and RNA-seq data of sheep ruminal epithelial cells, sheep esophageal epithelial cells and sheep hepatocyte cells with ruminant-specific conserved non-coding elements (RSCNEs), and then they determined the rumen-associated active RSCNEs. Further experiments showed that these active RSCNEs function as enhancers during rumen development and evolution [105]. In addition, these experiments provided strong evidence that enhancers can determine species diversity.

Changes in enhancer activity can affect phenotypic evolution in animals. Enhancers can change their activity during evolution by increasing and decreasing their number of TF binding sites, which in turn can affect the transcriptional regulatory network of genes and this can control the traits of organisms [106]. Cotney et al. compared the levels of the histone acetylation of enhancers affecting the development of limbs in human and rhesus monkeys, and they found that 11% of them gained activity in humans when compared to rhesus monkeys. This showed that the altered enhancer activity can be a molecular factor driving human limb evolution [107]. Human accelerated regions (HARs) are highly conserved genomic elements that are among the fastest evolving regions in the human genome, and nearly half of these play important roles in the developmental remodeling of the human brain [108]. Of these elements, nearly 49% were also found to be active enhancers of neurodevelopment [109]. Comparing HARs with enhancer activity to chimpanzee homologous regions, 68% of these elements showed different enhancer activity and the activity of HARs was higher than that seen in chimpanzees [87]. This demonstrates a close correlation between altered neurodevelopmental enhancer activity in HARs and human brain evolution.

Enhancer hijacking is an important cause of animal evolution. Chromatin conformational changes mediated by structural variants can trigger enhancer hijacking, which in turn reshapes the regulatory network of genes. This can cause genes to either gain or lose regulatory signals, thereby affecting gene expression. Keough et al. identified accelerated evolutionary regions in the genomes of humans and chimpanzees using 241 mammalian genome comparisons. Chromatin conformations of chimpanzee and human neural progenitor cells were assessed by chromatin conformation capture technology, and it was found that HARs were significantly enriched in TADs remodeled by human-specific structural variants (hsSVs) when compared to those in chimpanzees. This resulted in the differential expression of several genes [110]. This also revealed that enhancer hijacking may be responsible for the rapid evolution of HARs. Zhao et al. compared the structures of human and porcine TADs and found that the boundaries of 14 TADs were rearranged in humans. The expression levels of genes in the TADs in humans were significantly higher than those in the same genes in pigs within the same tissue. This finding suggests that the rearrangement of these TADs caused enhancer hijacking, which in turn, regulated gene expression between humans and pigs. Analysis of these rearrangements of TAD genes revealed that most of them were associated with human head and face phenotypes. Moreover, these genes were also demonstrated to be associated with phenotypic abnormalities of the head and face in other species, such as mice. These results suggest that enhancer hijacking caused by TAD rearrangements between pig and human chromosomes may have contributed to the evolution of head and face phenotypes [111].

The functions of enhancers are widely conserved in evolution. However, tissue-specific enhancer sequences are extremely poorly conserved in mammalian genomes, and sequence-similar enhancers are even rarer over large evolutionary distances [112]. Interestingly, despite sequence differences, enhancer functions are widely conserved across animal species distributed across the evolutionary tree. In mammals, Islet is involved in neurological and cardiac developmental functions as well as other functions. Islet enhancer sequences from sponges were shown to drive the cell-specific expression of developmental genes when inserted into both zebrafish and mice. This was seen even in cell types not found in sponges [113]. Functionally similar enhancer sequences are also not fully conserved, which stems from the stabilization of the binding TF core motif on the enhancer. These core motifs enable the enhancer to retain its own function while allowing it to generate new TF-binding motifs as well as the deletion of other motifs in order to complete evolution [113].

When a single deleterious mutation occurs in the genome, another mutation can occur in order to counteract the deleterious effects of the single mutation, and this is a phenomenon known as compensatory evolution. The process of compensatory evolution is widespread in enhancers [114]. When several enhancers work together to regulate a gene, one of them can become less active while another acts in its place to maintain the stability of gene expression [115]. This regulatory mechanism can accumulate genetic variations for evolution and it can also reduce the deleterious effects of mutations. Krüppel is a transcription factor that shapes the anterior–posterior axis of the embryo at the blastoderm stage. Several enhancers in *Drosophila melanogaster*, *D. yakuba* and *D. pseudoobscura* were found that could regulate Krüppel. These enhancers were able to act synergistically to conserve Krüppel expression levels among these three species, but they were unable to maintain this expression level of conservation when only individual enhancers were present [116]. Moreover, sequence analysis and experimental validation showed that these enhancers were all activated by different TFs. These results suggest that compensatory evolution has occurred between different enhancers that regulate the same gene.

### 7.3. CREs and Mammalian Diseases

EP interactions regulate the transcriptional expression mechanisms of genes and these can have a direct impact on tumorigenesis and development. Immune escape is an essential cause of tumorigenesis. The ability of PD-L1 on the surface of tumor cells to bind to PD-1 on the surface of T cells leads to a decrease in T cell activity, resulting in the immune escape of tumor cells [117]. By combining RNA-seq, Hi-C and ChIP-seq data, we found that there was an enhancer that significantly interacted with PD-L1 at 140 kb downstream of the sequence in tumor cells. In addition, when this enhancer was knocked down, the transcription and expression levels of PD-L1 were significantly reduced, which correspondingly inhibited the immune escape of cancer cells [118]. This suggests that alterations in the activity of some enhancers may be responsible for the development of cancer.

Many tumors arise and develop from changes in the location of CREs in DNA recombination. Extrachromosomal circular DNAs (eccDNAs), which are products of DNA recombination, are a manifestation of genomic instability, and they have highly open chromatin structures and are capable of freeing the EPs associated with them. EccDNAs are relatively scarce in normal cells and are highly enriched in tumor cells [119,120]. It has been found that oncogenes and their adjacent enhancers can be amplified and regulated by some eccDNAs, thus exerting a strong pro-cancer effect [119]. Enhancers play an important role in the oncogene-cyclic amplification-mediated pro-oncogenic effect, which breaks the pattern of gene regulation on chromosomes restricted by chromosomal conformation. This newly found interaction pattern between enhancers and oncogenes on the eecDNAs of ringed chromosomes reveals an important factor in tumor development and provides potential new ideas for effective tumor suppression.

Because of the spatiotemporal specificity of enhancers, the editing of enhancers usually does not affect the expression of their target genes in other cell types. Therefore, targeting therapy by using enhancers has often been the preferred type of gene therapy. In sickle cell anemia and β-thalassemia, targeting certain enhancers for therapy has been used in clinical practice. The GATA1 binding site on the *BCL11A* enhancer in a diseased mouse’s CD34 (+) cells was initially deleted by using gene editing techniques, and the targeted enhancer-edited cells were then infused back into the mouse [53,121]. This method specifically inhibited the expression of BCL11A in only red lineage cells, which prompted the expression of γ-globin and re-formed fetal hemoglobin, HbF, with α-globin. This partly restored the HbF level in the mouse, and the animal survived without needing further transfusion therapy. Compared with the treatment method of targeting genes, this technique has a great advantage to become a new target of gene editing in order to treat certain diseases.

## 8. Discussion

Performing functional research into CREs can be realized in three aspects, i.e., gene therapy, gene editing and gene expression regulation. Specifically, in gene therapy, CREs are used to regulate gene expression in specific tissues. For gene editing, the structure and function of CREs can be changed in order to achieve the precise editing of specific genes. For the regulation of gene expression, the specific mechanism of CRE action needs to be explored, and gene function and cellular development as well as other biological issues should be explored by combining CREs to regulate the gene expression levels.

In order to achieve the above applications, there are some problems in CREs that need to be addressed urgently. For example, there are great similarities in the sequence structures and functional characteristics of EPs, and as mentioned above, studies have shown that some of these molecules can act on each other, so it remains to be investigated whether a new definition can be established by combining the strengths and weaknesses of the TF capabilities. The most widely used high-throughput sequencing technology for detecting enhancers is based on the sequence characteristics of the promoters, rather than the enhancers themselves. Therefore, future research must be conducted on enhancer high-throughput sequencing technology, as to whether it can focus on eRNAs, so as to more accurately identify enhancers that need to be explored. As for the functional study of enhancers, it is important not to consider only single enhancers but also to take into account the redundancy of some of these molecules and to consider that collaboration between them may occur to functionally compensate for their absence of individuality.

In recent years, the link between transposable elements (TEs) and CREs has been gradually revealed. TEs account for about half of the mammalian genome, and about 20% of the mammalian CREs are derived from TEs [122]. Transposons can be individually replicated and cleaved from the genome, cyclized and inserted into other loci, and TEs can be silenced by host genes and accumulate mutations that no longer propagate while retaining their own functional properties [123]. In addition, many TEs have several TF binding sites, function as enhancers and some transposons are able to create new host gene transcripts by introducing promoters, which is often associated with animal evolution and disease occurrence [124]. TEs appears to be an indispensable part of the future study of CREs as they open up new directions for scientists to study CREs. In addition, in the study of CREs and development, evolution and disease, one cannot only focus on the function and activity of their regulatory element but should also introduce the concepts of time and changes in the spatial structure of the chromatin ultrastructure (e.g., changes in TAD and the production of eccDNAs). This adds a temporal dimension to the spatial configuration of the molecules. This will allow us to reveal the spatio-temporal specific changes in the function and activity of regulatory elements and enrich the regulatory network of genes, which is not only the ultimate goal of future studies of CREs but is also the beginning of our entry into four-dimensional genomics.

## Figures and Tables

**Figure 1 ijms-25-00343-f001:**
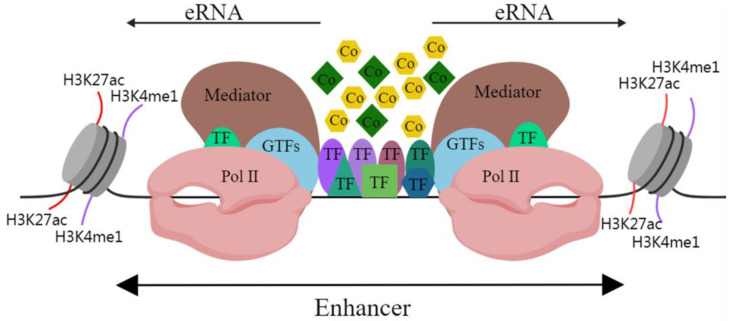
Schematic diagram of enhancer structure. Abbreviations: H3K27ac, acetylation of lysine 27 on histone 3; H3K4me1, monomethylation on lysine 4 of histone H3; TF, transcription factor; GTF, general transcription factor; Pol II, RNA polymerase II; Co, cofactor. The black line indicates DNA.

**Figure 2 ijms-25-00343-f002:**
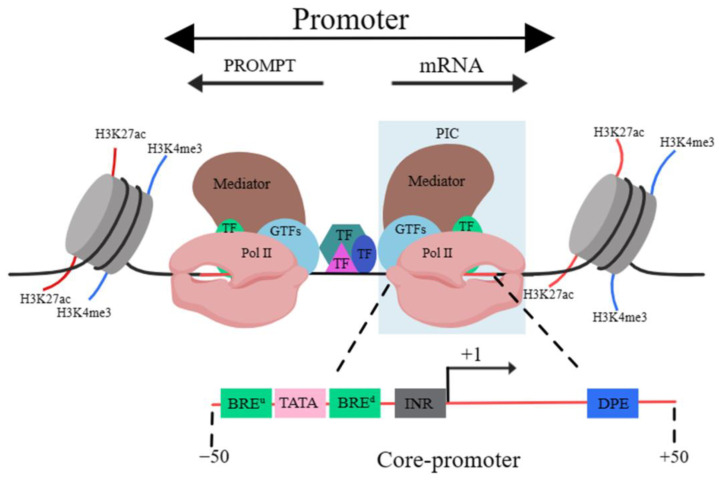
Schematic diagram of promoter structure. Abbreviations: H3K27ac, acetylation of lysine 27 on histone 3; H3K4me3, Trimethylation of histone H3 lysine 4. TF, transcription factor; GTF, general transcription factor; Pol II, RNA polymerase II; PIC, pre-initiation complex; BRE^u^, BRE upstream (of TATA box); BRE^d^, BRE downstream (of TATA box); Inr, initiator element; DPE, downstream promoter element. The black line indicates DNA.

**Figure 3 ijms-25-00343-f003:**
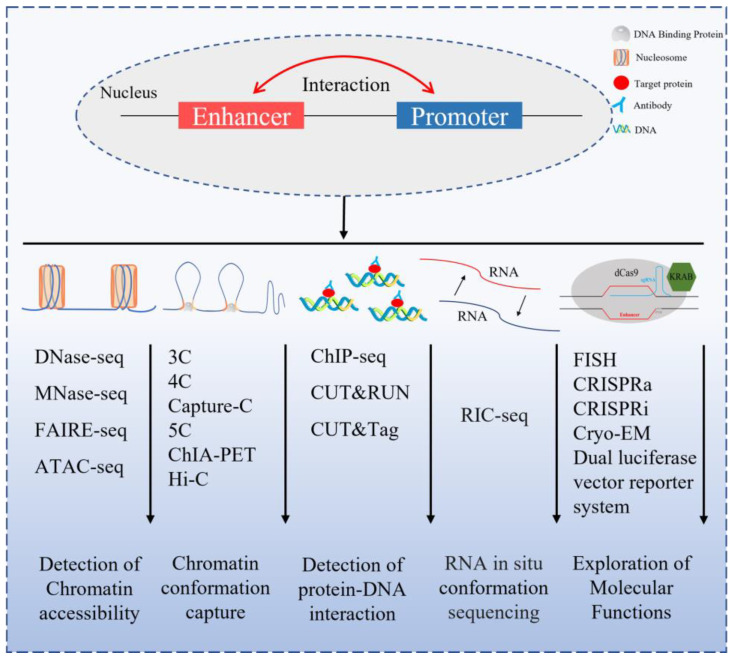
A summary diagram of the techniques used for studying the EP interactions.

**Figure 4 ijms-25-00343-f004:**
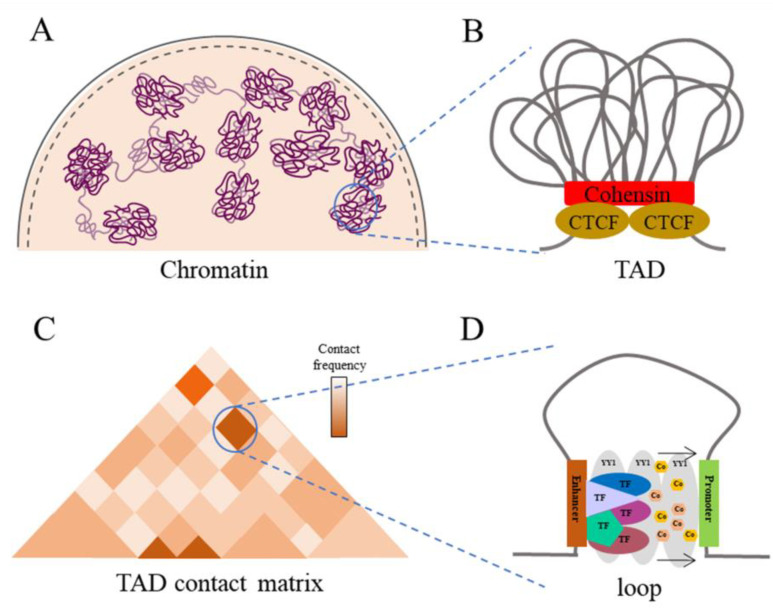
Schematic diagram of the chromatin loop structure. (**A**) Morphological diagram of chromatin in the nucleus. (**B**) A TAD structure diagram. Abbreviations: TAD, topologically associated structural domains; Cohesin, adhesion protein; CTCF, CCCTC-binding factor. (**C**) A TAD contact matrix diagram. (**D**) Diagram of the loop structure. Abbreviations: TF, transcription factor; YY1, transcription factor Yin Yang 1; Co, cofactor. Gray lines indicate DNA.

**Table 1 ijms-25-00343-t001:** Identification techniques for enhancers and promoters.

Assay	Full Assay Name	Theory	Application	Advantage	Shortcomings
ChIP-seq [21]	Chromatin immunoprecipitation followed by sequencing.	Formaldehyde crosslinking; fragmented DNA; antigen-antibody specific recognition; DNA purification; DNA amplification; sequencing.	Genome-wide identification of enhancers and promoters by histone modification of DNA.	Broad application.	Formaldehyde cross-linking can lead to non-specific binding: genome fragmentation can produce inhomogeneous fragments, all of which can affect the final identification results.
CUT&RUN [22]	Cleavage under target and release using nuclease.	Cell permeabilization; antigen-antibody specific recognition; pAG-MNase digestion; DNA purification; DNA amplification; sequencing.	Genome-wide identification of enhancers and promoters by histone modification of DNA.	No cross-linking or DNA breakage required; low cell count requirements; high signal-to-noise ratio; accurate results.	Cumbersome steps to build a library; some antibodies may not function under non-cross-linked conditions, thus limiting the scope of application.
CUT&Tag [23]	Cleavage under targets and tagmentation.	Cell permeabilization; antigen-antibody specific recognition; pA-Tn5 digestion; DNA purification; DNA amplification; sequencing.	Genome-wide identification of enhancers and promoters by histone modification of DNA.	Eliminates the need for cross-linking and DNA fragmentation; simplifies library building steps; has the highest signal-to-noise ratio and experimental efficiency.	Some antibodies may not function under non-cross-linked conditions, thus limiting the scope of application.
DNase-seq [24]	Deoxyribonuclease I (DNase I)-hypersensitive site sequencing.	Chromatin fragmentation; DNase I digestion; DNA purification; DNA amplification; sequencing.	Genome-wide identification of enhancers and promoters by identifying chromatin accessibility.	Simplicity of experimental operation; wide range of applications.	High cell count requirements; difficulty in controlling the digestion time; large errors in sequencing results.
MNase-seq [25]	Micrococcal nuclease (MNase) sequencing.	Formaldehyde crosslinking; excess MNase (with both endonuclease and exonuclease activity) digestion; DNA purification; sequencing.	Genome-wide identification of enhancers and promoters by identifying chromatin accessibility.	In contrast to conventional chromatin accessibility detection techniques, digestion of open chromatin regions allows for the backward inference of the region of chromatin accessibility.	High cell count requirements; MNase has sequence preference on the genome.
FAIRE-seq [26]	Formaldehyde-assisted isolation of regulatory elements sequencing.	Chromatin fragmentation; formaldehyde crosslinking; solubility differences; liquid phase separation; DNA purification sequencing.	Genome-wide identification of enhancers and promoters by identifying chromatin accessibility.	Directly enriched regions of open chromatin; broad application; no sequence preference.	High dependence on formaldehyde fixation efficiency; too low signal-to-noise ratio.
ATAC-seq [27]	Assay for transposase-accessible chromatin with high-throughput sequencing.	Nucleoplasmic separation; chromatin fragmentation; DNA purification; DNA amplification; sequencing.	Genome-wide identification of enhancers and promoters by identifying chromatin accessibility.	Simplifies library building steps; low cell demand; simplicity of experimental operation.	The optimum amount required for different cells may vary.
GRO-seq [28]	Global run-on sequencing.	Cell nucleus extraction; Br-UTP labeling; RNA purification; reverse transcription; sequencing.	Genome-wide identification of enhancers and promoters by identifying transcription products.	Maps position of transcriptionally engaged RNA polymerases; detects transcription anywhere on the genome; provides robust coverage of enhancer- and promoter-associated RNAs.	Artifacts may be introduced during the preparation of the nuclei; resolution is only 30-100 nt; requires nascent RNAs of at least 18 nt.
PRO-seq [29]	Precision nuclear run-on sequencing.	Cell nucleus extraction; biotin labeling; RNA purification; reverse transcription; sequencing.	Genome-wide identification of enhancers and promoters by identifying transcription products.	High sensitivity; single base pair resolution can be achieved; high signal-to-noise ratio; short and unstable RNAs can be identified.	Unable to detect arrested or backtracked RNAPII complexes.
CAGE-seq [30]	Cap analysis of gene expression sequence.	RNA extraction; reverse transcription; biotin labeling; cDNA purification; sequencing.	Genome-wide identification of enhancers and promoters by identifying transcription products.	High sensitivity; high accuracy.	Only works on total mature RNA.
Start-seq [31]	Small 5′-capped RNA sequencing.	Cell nucleus extraction; RNA extraction; exonuclease digestion; RNA purification; reverse transcription; cDNA amplification; sequencing.	Genome-wide identification of enhancers and promoters by identifying transcription products.	High sensitivity; high specificity; high signal-to-noise ratio.	Cumbersome experimental steps.
SIF-seq [32]	Site-specific integration fluorescence-activated cell sorting followed by sequencing.	DNA fragmentation; vector construction; fluorescence-activated cell sorting; DNA amplification; sequencing.	Identify enhancers and promoters by recognizing the activity of regulatory elements.	High precision; ability to integrate into the genome for repeat identification of regulatory element activity.	Inability to perform high-throughput assays; vulnerable to cell specificity.
MPRA [33]	Massively parallel reporter assay.	DNA fragmentation; barcode tags; vector construction; RNA extraction; reverse transcription; cDNA amplification; sequencing.	Identify enhancers and promoters by recognizing the activity of regulatory elements.	Wide range of applications.	Inability to perform high-throughput assays; cumbersome experimental steps; episomal reports may not accurately reflect the function of the regulatory element in its endogenous context.
STARR-seq [34]	Self-transcribing active regulatory region sequencing.	DNA fragmentation; vector construction; RNA extraction; reverse transcription; cDNA amplification; sequencing.	Genome-wide identification of enhancers and promoters by identifying regulatory element activity.	Provides high-throughput promoter and enhancer detection; simplicity of experimental operation.	Episomal reports may not accurately reflect the function of the regulatory element in its endogenous context.

**Table 2 ijms-25-00343-t002:** Summary of the conserved cis-regulatory elements found in mammals.

Type	Examples of Major Species	Links	References
Ruminant	Cattle, sheep, goats	http://animal.omics.pro/code/index.php/main (accessed on 15 October 2023)	[46]
Vertebrate	Human, mouse, rat	http://hgdownload.cse.ucsc.edu/goldenPath/hg18/multiz28way (accessed on 15 October 2023)	[47]
Primate	Humans, gibbons, rhesus monkeys	https://www.science.org/doi/suppl/10.1126/sciadv.adc9507/suppl_file/sciadv.adc9507_data_s1_to_s6.zip (accessed on 15 October 2023)	[45]

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
