# Peer review of "Cis-Regulatory Elements in Mammals"

_ijms, 2023, doi:10.3390/ijms25010343_

Round 1
Reviewer 1 Report
Comments and Suggestions for Authors
The review by Liu et al describes the typical characteristics and role of enhancers and promoters in the regulation of gene expression in mammals. The review is interesting, clearly written and well structured.
Major comments:
1. The authors discuss that many key characteristics of promoters and enhancers are similar. It is true. However, Figures 1 and 2 are very different:
- Figure 1 shows bidirectional expression in the enhancer, while Figure 2 shows only unidirectional expression.
- To improve perception, it is desirable to depict growing RNA chains in both figures (for both directions).
2. CAGE-seq allows not only to recognize the position of enhancers, but also to evaluate their activation using eRNA expression, i.e. in fact, enhancer activity. It means that the division into paragraphs 3.5 and 3.6 is very arbitrary.
Minor comments:
L 59. Please, explain acronym HS2.
L 61-63. It would be appropriate to point out here that the mechanisms of regulation of target gene expression by eRNA will be discussed in more detail below.
L 88 replace “When” with “when”.
L 103-107 It is advisable to point out that about 10% of all promoters are bidirectional promoters.
Author Response
- The authors discuss that many key characteristics of promoters and enhancers are similar. It is true. However, Figures 1 and 2 are very different:
- Figure 1 shows bidirectional expression in the enhancer, while Figure 2 shows only unidirectional expression.
- To improve perception, it is desirable to depict growing RNA chains in both figures (for both directions).
Response:Thank you for your suggestion. We have redrawn Figure 2 and highlighted promoter bidirectional transcription in the figure. The enhancers and promoters depicted in Figures 1 and 2 are in the accumulation state, and therefore the growing RNA strand cannot be accurately mapped. However, the direction of RNA generation is indicated in the figures for ease of understanding.
- CAGE-seq allows not only to recognize the position of enhancers, but also to evaluate their activation using eRNA expression, i.e. in fact, enhancer activity. It means that the division into paragraphs 3.5 and 3.6 is very arbitrary.
Response:Thank you for your suggestion. We agree with you that eRNA can in some way indicate that the enhancer is active. However, in paragraphs 3.5 we wanted to summarize methods for identifying enhancers based solely on transcription products. In paragraph 3.6, as mentioned at the beginning of the paragraph, we would like to summarize the methods that can directly test the activity of enhancers, e.g., the luciferase reporter system that can directly verify the activity of enhancers directly by the intensity of the bioluminescence.
Minor comments:
- L 59. Please, explain acronym HS2.
Response:Thank you for your suggestion. Human β-like globin genes are organized in the order of 5′-ε-Gγ-Aγ-δ-β-3′ within the β-globin locus. The expression of β-like globin genes is regulated by 3′HS1 and five DNase I hypersensitive sites (5′HS5~5′HS1) in a locus control region. The HS2 enhancer transcribes enhancer RNA and regulates the expression of ε-globin, γ-globin and β-globin. In addition, in response to your suggestion, we have made a brief addition to HS2 in the manuscript.
- L 61-63. It would be appropriate to point out here that the mechanisms of regulation of target gene expression by eRNA will be discussed in more detail below.
Response:Thank you for your comments, we have added relevant statements in the manuscript to lead into the mechanisms by which eRNAs regulate target genes that will be explored below.
- L 88 replace “When” with “when”.
Response:Thank you for your careful advice. We have made changes.
- L 103-107 It is advisable to point out that about 10% of all promoters are bidirectional promoters.
Response:This is a good question. We have consulted the relevant information. The promoters we mentioned in the manuscript are the promoters of most genes. These promoters produce mRNA and PROMPT during bi-directional transcription, whereas bi-directional promoters produce two mRNAs during bi-directional transcription. so these are essentially two different promoters.
Reviewer 2 Report
Comments and Suggestions for Authors
This paper is well-constructed, lucidly written, has appropriate references, and fine figures, thus representing a valuable addition to the literature. I would recommend publication.
Minor notes:
line 53 - The first mention of Pol II should be 'RNA polymerase II (Pol II)'. This is included in Fig. 1 but Pol II appears earlier in the body of the section.
Similarly, line 54: eRNA should be 'enhancer RNA (eRNA).'
line 424: change 'mutation' to 'a mutation within' or 'mutating'
Author Response
Minor notes:
1、line 53 - The first mention of Pol II should be 'RNA polymerase II (Pol II)'. This is included in Fig. 1 but Pol II appears earlier in the body of the section.
Response:Thank you for your suggestion. We have revised Pol II to RNA polymerase II (Pol II) where it first appears in the manuscript.
2、Similarly, line 54: eRNA should be 'enhancer RNA (eRNA).'
Response:Thank you for your careful comments, we have revised eRNA to enhancer RNA (eRNA) where it first appears in the manuscript.
3、line 424: change 'mutation' to 'a mutation within' or 'mutating'
Response:Thank you for your suggestion. We changed the mutation at the corresponding position to mutating.